# TRAIN SHORT, TEST LONG IN COMBINATORIAL OPTIMIZATION

## ABSTRACT

The inherent characteristics of the Transformer enable us to train on shorter datasets and extrapolate to testing on longer ones directly. Numerous researchers in the realm of natural language processing have proposed a variety of methods for length extrapolation, the majority of which involve position embeddings. Nonetheless, in combinatorial optimization problems, Transformers are devoid of position embeddings. We aspire to achieve successful length extrapolation in combinatorial optimization problems as well. As such, we propose an entropy invariant extrapolation method (EIE), which obviates the need for positional embeddings and employs varying scale factors according to different lengths. Our approach eliminates the need for retraining, setting it apart from prior work. Results on multiple combinatorial optimization datasets demonstrate that our method surpasses existing ones.

## 1 INTRODUCTION

The intersection of machine learning and combinatorial optimization has been established since early times, with numerous combinatorial problems involving sequential decision-making. For instance, the Traveling Salesman Problem (TSP) pertains to the sequence of site visits, while the offline bin packing problem concerns the order of selecting boxes. These sequential decision-making scenarios can be approximated as Markov Decision Processes within the realm of Reinforcement Learning. As early as the 1980s, Hopfield & Tank (1985) deployed neural networks to solve small-scale TSP instances. Recently, with the advent of computational advancements, there has been a resurgence in employing machine learning for combinatorial optimization. Vinyals et al. (2015) utilized the Pointer Network for solving the TSP. However, the Pointer Network is not permutation equivariant , meaning that shuffling the input sequence of sites alters the output sequence, casting doubt on the credibility of the results. A key module within Transformers (Vaswani et al., 2017), known as self-attention, is permutation equivariant when devoid of positional embeddings (Lee et al., 2019). Consequently, this led to the introduction of self-attention for tackling combinatorial optimization problems (Kool et al., 2019; Zhang et al., 2021; Jiang et al., 2021). Simultaneously, another advantage of self-attention is that once a Transformer is trained on a shorter dataset, it can be directly deployed for inference on longer datasets without necessitating changes to the network architecture. This characteristic has spurred research endeavors to augment the length extrapolation competencies of Transformers. Achieving satisfactory extrapolation properties could considerably curtail the training duration.

The attention matrix of a Transformer represents the correlation between different tokens. Since each row of the matrix sums to 1, and each element is non-negative, we can interpret each element as a probability and analyze it using entropy. Our experimental results indicate that under certain conditions, the smaller the relative rate of entropy change, the better the extrapolation performance of the Transformer. Therefore, we introduce a method, dubbed entropy invariant extrapolation (**EIE**), that assigns distinct scale factors to different sequence lengths, aiming to equalize their expected entropy as closely as possible.

## 2 RELATED WORK

Research on Transformer-based length extrapolation can be bifurcated into two categories: those that involve position embeddings(IPO) and those that do not involve position embeddings(NPO) .

**IPO** A substantial proportion of research has capitalized on the prior linguistic knowledge that closer tokens are more closely related, leading to the design of various position embeddings (Vaswani et al., 2017; Press et al., 2021; Chi et al., 2022; 2023; Sun et al., 2022; Su, 2023b;d;e;f; Su et al., 2021). (Ruoss et al., 2023) put forth randomized position embeddings that simulate positions for elongated sequences and randomly select an ordered subset to accommodate the sequence length. The conventional Transformer does not incorporate a bias term in the computation of Query and Key in multi-head attention (Vaswani et al., 2017). Nevertheless, Su (2023a) found that incorporating a bias term when computing Query and Key with rotary position encoding(RoPE) (Su et al., 2021) can significantly improve extrapolation performance. Chen et al. (2023) introduced position interpolation, which proportionately compresses the length in position embeddings, extrapolates the length to 32,768, and fine-tunes the model for 1000 steps. bloc97 (2023) demonstrated the NTK-aware Scaled RoPE, which leverages interpolation for low frequencies and direct extrapolation for high frequencies in Sinusoidal position embeddings, yielding satisfactory results without the necessity for fine-tuning. Su (2023c) provided an understanding of NTK-aware Scaled RoPE from the radix system perspective and generalized NTK-aware Scaled RoPE. Dubois et al. (2020); Ray Chowdhury & Caragea (2023) incorporated location attention. Bueno et al. (2022) discovered that generating step-by-step rationales and introducing marker tokens can enhance the model's extrapolation capability without altering the structure of the large language model. Chu et al. (2023) proposed a learnable position embedding designed specifically for visual Transformers (Dosovitskiy et al., 2021; Touvron et al., 2021). Kazemnejad et al. (2023) asserted that explicit position encodings are not a prerequisite for decoder-only Transformers to effectively generalize to longer sequences.

**NPO** In the context of standard multi-head attention, the scaling factor is recognized as $\frac{1}{\sqrt{d}}$, where $d$ represents the head size. Chiang & Cholak (2022) introduced a scaling factor of $\frac{\ln n}{\sqrt{d}}$, with $n$ being the sequence length. Conversely, Su (2021) employed a scaling factor of $\frac{\ln n}{d}$.

## 3 BACKGROUND

### 3.1 PERMUTATION EQUIVARIANT

Let's denote $S_n$ as the symmetric group on $n$ letters. For $\sigma \in S_n$, we define $\sigma(\boldsymbol{X}) = (\boldsymbol{X}_{\sigma(1),:}^T, \boldsymbol{X}_{\sigma(2),:}^T, ..., \boldsymbol{X}_{\sigma(n),:}^T)$, where $\boldsymbol{X} \in \mathbb{R}^{n \times d_1}$, $d_1$ can be any positive integer, and $\boldsymbol{X}_{i,:} \in \mathbb{R}^{d_1}$ indicates the $i$-th row vector of $X$. We now give the definition of permutation equivariant .

**Definition 1 (permutation equivariant )** $f : \mathbb{R}^{n \times d_1} \to \mathbb{R}^{n \times d_2}$ *is a mapping. If* $\forall \boldsymbol{X} \in \mathbb{R}^{n \times d_1}$ *and* $\forall \sigma \in S_n$, *it holds that* $f(\sigma(\boldsymbol{X})) = \sigma(f(\boldsymbol{X}))$, *then* $f$ *is permutation equivariant .*

To distinguish from the standard Transformer (Vaswani et al., 2017), we denote the Transformer utilized in combinatorial optimization as the **CO-Transformer**. We represent the CO-Transformer as $f_{co} = f_{co,1} \circ f_{co,2} \circ ... \circ f_{co,m}$, where each $f_{co,i} : \mathbb{R}^{n \times d_i} \to \mathbb{R}^{n \times d_{i+1}}$ signifies the $i_{th}$ layer of the CO-Transformer. Typically, this would be one of the following: multi-head self-attention without position embeddings (Vaswani et al., 2017), fully connected layers, LayerNorm (Ba et al., 2016), or Softmax. Furthermore, we have $d_{m+1} = 1$.

**Theorem 1** *CO-Transformer is permutation equivariant .*

Why can't CO-Transformers add positional embeddings in combinatorial optimization problems? Intuitively, the inputs to the CO-Transformer in combinatorial optimization problems do not have a temporal order like time series and language. Mathematically, a CO-Transformer with positional embeddings will not be permutation equivariant . In combinatorial optimization problems, if the network is not permutation equivariant , some illogical situations will arise. For instance, in the TSP, the input to the neural network is the coordinates of the locations, and each component of the network's output represents the probability of going to that location next. If the network is not permutation equivariant , when we shuffle the order of the locations in the input, the probabilities of going to each location in the output will change, which is illogical.

### 3.2 BASIS FOR EXTRAPOLATING LENGTH

CO-Transformer is composed of self-attention without position embeddings , fully connected layers, Softmax and LayerNorm (Ba et al., 2016). In solving combinatorial optimization problems with deep reinforcement learning, the attention layer is typically a multi-head self-attention(Vaswani et al., 2017). We denote the input to the multi-head self-attention as $\boldsymbol{X} \in \mathbb{R}^{n \times (h \times d)}$, where $n$ is the sequence length, $h$ indicates the number of heads, and $d$ represents the size of each head. The computational process of multi-head self-attention is as follows:

$$\boldsymbol{Q}_i = \boldsymbol{X}\boldsymbol{W}_Q^i, \boldsymbol{K}_i = \boldsymbol{X}\boldsymbol{W}_K^i, \boldsymbol{V}_i = \boldsymbol{X}\boldsymbol{W}_V^i, \quad i = 1, 2, ..., h \tag{1}$$

where $\boldsymbol{W}_Q^i, \boldsymbol{W}_K^i, \boldsymbol{W}_V^i \in \mathbb{R}^{(h \times d) \times d}$.

$$\boldsymbol{A}_i = Softmax(\lambda \boldsymbol{Q}_i \boldsymbol{K}_i^T) \in \mathbb{R}^{n \times n}, \quad i = 1, 2, ..., h \tag{2}$$

where $\lambda$ is scaling factor, $\lambda = \frac{1}{\sqrt{d}}$ and $\boldsymbol{K}_i^T$ is the transpose of $\boldsymbol{K}_i$.

$$\boldsymbol{O}_i = \boldsymbol{A}_i \boldsymbol{V}_i \in \mathbb{R}^{n \times d}, \quad i = 1, 2, ..., h \tag{3}$$

$$\boldsymbol{O} = concat(\boldsymbol{O}_1, \boldsymbol{O}_2, ..., \boldsymbol{O}_h)\boldsymbol{W}_O \in \mathbb{R}^{n \times (h \times d)} \tag{4}$$

where $\boldsymbol{W}_O \in \mathbb{R}^{(h \times d) \times (h \times d)}$.

It is easy to note that $\boldsymbol{W}_O, \boldsymbol{W}_Q^i, \boldsymbol{W}_K^i, \boldsymbol{W}_V^i, \quad i = 1, 2, ..., h$ are independent of $n$. Thus, by using inputs $\boldsymbol{X}$ of varying lengths $n$, we can still obtain outputs when passing them through the same multi-head self-attention layer. The same conclusion holds for fully connected layers, residual connections, and LayerNorm. This is the basis for the Transformer's ability to extrapolate length.

## 4 METHOD

Our method, EIE, consists of two stages. In the first stage, we utilize a neural network to compute distinct scaling factors for different sequence lengths. In the second stage, we substitute the original scaling factor in the Transformer with the newly computed scaling factor from the first stage. We will now delve into the motivation behind our method, followed by a detailed exposition of our procedure and neural network architecture.

### 4.1 ANALYSIS OF ENTROPY

Entropy is a measure of uncertainty, which can also be perceived as the degree of focus in attention: Taking the bin packing problem as an example, it can be envisioned as placing several boxes into a bin. In this scenario, the input to the neural network consists of the features of all the boxes. When entropy is zero, attention is concentrated on a single box. If the entropy is $\ln n$, then the attention is evenly distributed across all boxes. The introduction of new boxes increases entropy, and a well-trained Transformer is accustomed to a certain range of entropy. Therefore, we strive to maintain a consistent level of entropy during extrapolation. Specifically, our goal is to sustain attention on an equivalent number of boxes, preventing attention from becoming overly diluted even when new boxes are introduced, thereby significantly altering the overall sum. We use $\boldsymbol{Q}, \boldsymbol{K}, \boldsymbol{A}$ to replace $\boldsymbol{Q}_i, \boldsymbol{K}_i, \boldsymbol{A}_i$ in Equations 1 and 2. $\boldsymbol{Q}_{i,:}, \boldsymbol{K}_{i,:}$ represent the i-th row vector of $\boldsymbol{Q}, \boldsymbol{K}$. As every row sum of $\boldsymbol{A}$ is 1 and each element is non-negative, we can analyze it using entropy, $H_i$ represents the entropy of the $i$-th row in $\boldsymbol{A}$.

$$H_i = -\sum_{j=1}^{n} \frac{e^{\lambda \boldsymbol{Q}_{i,:} \boldsymbol{K}_{j,:}^T}}{\sum_{j=1}^{n} e^{\lambda \boldsymbol{Q}_{i,:} \boldsymbol{K}_{j,:}^T}} \ln \frac{e^{\lambda \boldsymbol{Q}_{i,:} \boldsymbol{K}_{j,:}^T}}{\sum_{j=1}^{n} e^{\lambda \boldsymbol{Q}_{i,:} \boldsymbol{K}_{j,:}^T}}$$

$$= \sum_{j=1}^{n} \frac{e^{\lambda \boldsymbol{Q}_{i,:} \boldsymbol{K}_{j,:}^T}}{\sum_{j=1}^{n} e^{\lambda \boldsymbol{Q}_{i,:} \boldsymbol{K}_{j,:}^T}} (\ln \sum_{j=1}^{n} e^{\lambda \boldsymbol{Q}_{i,:} \boldsymbol{K}_{j,:}^T} - \ln e^{\lambda \boldsymbol{Q}_{i,:} \boldsymbol{K}_{j,:}^T}) \tag{5}$$

$$= \ln \sum_{j=1}^{n} e^{\lambda \boldsymbol{Q}_{i,:} \boldsymbol{K}_{j,:}^T} - \frac{\sum_{j=1}^{n} \lambda \boldsymbol{Q}_{i,:} \boldsymbol{K}_{j,:}^T e^{\lambda \boldsymbol{Q}_{i,:} \boldsymbol{K}_{j,:}^T}}{\sum_{j=1}^{n} e^{\lambda \boldsymbol{Q}_{i,:} \boldsymbol{K}_{j,:}^T}}$$

Next, we analyze the distribution of $\boldsymbol{Q}_{i,:}, \boldsymbol{K}_{j,:}$ in Equation 5. In Equation 1, every element of $\boldsymbol{X}$ can be seen as adhering to a distribution with a mean of 0 and a standard deviation of 1, primarily because the layer above multi-head attention is typically LayerNorm. We hypothesize that each element of $\boldsymbol{X}$ follows a normal distribution. If we employ Xavier initialization (Glorot & Bengio, 2010) for $\boldsymbol{W}_Q^i, \boldsymbol{W}_K^i$ in Equation 1, then according to the principles of Xavier initialization, the means and variances of the input and output are preserved. Consequently, each element in $\boldsymbol{Q}_{i,:}, \boldsymbol{K}_{j,:}$ also adheres to a standard normal distribution. Simultaneously, presuming $\boldsymbol{Q}_{i,:}, \boldsymbol{K}_{j,:}$ to be independent, we will discuss the rationale behind these assumptions in the appendix. Based on the above analysis, expected value of every row of $\boldsymbol{A}$ obeys the same distribution in terms of entropy. We denote the expected entropy of each row of $\boldsymbol{A}$ as $H(n, \lambda(n))$. The reason $\lambda(n)$ appears is that when we extrapolate, $d$ is fixed as we do not change the network structure, and so can be considered as a constant. Therefore, we have:

$$H(n, \lambda(n)) = \mathbb{E}_{\boldsymbol{Q}_{i,:}, \boldsymbol{K}_{j,:} \sim \mathcal{N}(\boldsymbol{0}, \boldsymbol{I}_d)} [\ln \sum_{j=1}^{n} e^{\lambda \boldsymbol{Q}_{i,:} \boldsymbol{K}_{j,:}^T} - \frac{\sum_{j=1}^{n} \lambda \boldsymbol{Q}_{i,:} \boldsymbol{K}_{j,:}^T e^{\lambda \boldsymbol{Q}_{i,:} \boldsymbol{K}_{j,:}^T}}{\sum_{j=1}^{n} e^{\lambda \boldsymbol{Q}_{i,:} \boldsymbol{K}_{j,:}^T}}] \tag{6}$$

With Equation 6, we can compute the expected entropy using the Monte Carlo method. We use $N$ to denote the sequence length during training and $N_{max}$ to denote the maximum sequence length during extrapolation. We hope that

$$\forall n \in \{N+1, N+2, ..., N_{max}\}, H(n, \lambda(n)) \approx H(N, \lambda(N)) \tag{7}$$

First, we analyze what properties $\lambda$ needs to satisfy to better meet Equation 7. As $n$ increases, the introduction of new boxes causes attention to scatter, leading to an increase in uncertainty. Only when $\lambda$ increases, the gap between $\lambda \boldsymbol{Q}_{i,:} \boldsymbol{K}_{j,:}^T$ will expand, which reduces uncertainty after the Softmax operation. Only by offsetting the two can entropy remain unchanged. This indicates that:

$$\frac{d\lambda}{dn} > 0 \tag{8}$$

Although $n$ is a positive integer, in Equation 8, we can extend $n$ to the real number field $\mathbb{R}$. We use the neural network $f(n; \boldsymbol{\theta})$ to approximate $\frac{d\lambda}{dn}$. The reason we do not use a neural network to approximate $\lambda$ is that if we use a neural network to approximate $\lambda$, it is difficult to ensure that Equation 8 strictly holds. However, if we use a neural network to approximate $\frac{d\lambda}{dn}$, we only need to apply a non-negative function to the output of the last fully connected layer of the neural network, and then add a tiny positive number to strictly satisfy Equation 8.

## 4.2 NETWORK ARCHITECTURE

Since the input and output of the neural network are both one-dimensional, we do not need to use a very complex network structure to achieve a good approximation. The neural network only contains fully connected layers and residual connections. We use $d'$ to denote the hidden size of the network. The network architecture is illustrated in Figure 1. The first layer is a fully connected layer with an activation function.

$$\boldsymbol{Y}^{(1)} = ReQUr(n\boldsymbol{W}^{(1)} + \boldsymbol{b}^{(1)}) \tag{9}$$

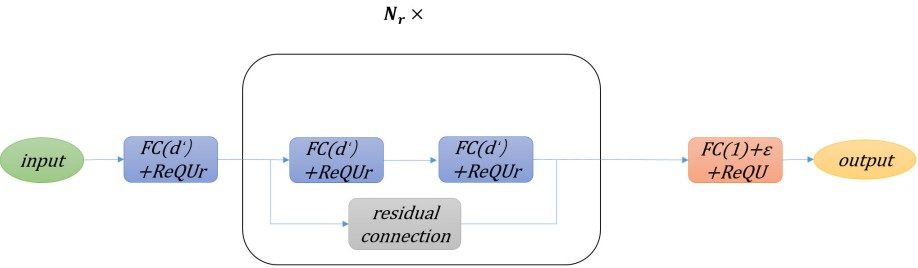

Figure 1: Network Architecture. In this figure, $FC(d')$ represents the fully connected layer with an output dimension of $d'$.

where $\boldsymbol{W}^{(1)} \in \mathbb{R}^{1 \times d'}, \boldsymbol{b}^{(1)} \in \mathbb{R}^{d'}$. The function $ReQUr(x) = ReLU^2(x) - ReLU^2(x - 0.5)$ is applied. Compared with $ReLU$, $ReQUr$ (Yu et al., 2021) performs better in approximating smooth functions. Next are $N_r$ residual blocks:

$$\boldsymbol{Y}^{(j)} = ReQUr\left(ReQUr\left(\boldsymbol{Y}^{(j-1)}\boldsymbol{W}^{(j,1)} + \boldsymbol{b}^{(j,1)}\right)\boldsymbol{W}^{(j,2)} + \boldsymbol{b}^{(j,2)}\right) + \boldsymbol{Y}^{(j-1)} \quad (10)$$

where $j = 2, 3, ..., N_r + 1$, $\boldsymbol{W}^{(j,1)}, \boldsymbol{W}^{(j,2)} \in \mathbb{R}^{d' \times d'}$, $\boldsymbol{b}^{(j,1)}, \boldsymbol{b}^{(j,2)} \in \mathbb{R}^{d'}$. The last layer is:

$$f(n; \boldsymbol{\theta}) = ReQU(\boldsymbol{Y}^{(N_r+1)}\boldsymbol{W}^{(N_r+2)} + \boldsymbol{b}^{(N_r+2)}) + \epsilon \quad (11)$$

where $\boldsymbol{W}^{(N_r+2)} \in \mathbb{R}^{d' \times 1}$, $\boldsymbol{b}^{(N_r+2)} \in \mathbb{R}^1$, $ReQU(x) = ReLU^2(x)$, and $\epsilon = 10^{-4}$. Compared with $ReLU$, $ReQU$ (Li et al., 2019) performs better in approximating smooth functions, but replacing the previous $ReQUr$ with $ReQU$ would increase the training difficulty.

$$\lambda(n) \approx \lambda(N) + \int_N^n f(x; \boldsymbol{\theta})dx \quad (12)$$

For the integral present in Equation 12, we employ a five-point Gaussian quadrature for its approximation, as demonstrated in Algorithm 1. The study by Su (2021) necessitates the Transformer model to be retrained, which can be time-consuming. We aim to circumvent the need to retrain the Transformer. Given that $\lambda = \frac{1}{\sqrt{d}}$ is utilized in many existing well-trained models, we establish $\lambda(N) = \frac{1}{\sqrt{d}}$ in Equation 12. Next, we provide the formulation of the loss function.

$$L = \frac{1}{B}\sum_{i=1}^{B}\left(H(n_i, \lambda_i) - H(N, \frac{1}{\sqrt{d}})\right)^2 \quad (13)$$

Where $B$ represents the batch size, $n_i$ signifies the integer sampled that is greater than $N$ but less than or equal to $N_{max}$, and $\lambda_i = \lambda(n_i)$. The training procedure can be seen in Algorithm 2.

---

**Algorithm 1** Five-Point Gaussian Quadrature$(a, b, f)$

---

**Input:** Integration start point $a$, end point $b$, integrand $f(x)$.

1: $x_1 = 0$, $x_2 = \frac{1}{3}\sqrt{5 - 2\sqrt{\frac{10}{7}}}$, $x_3 = -\frac{1}{3}\sqrt{5 - 2\sqrt{\frac{10}{7}}}$, $x_4 = \frac{1}{3}\sqrt{5 + 2\sqrt{\frac{10}{7}}}$, $x_5 =$

$-\frac{1}{3}\sqrt{5 + 2\sqrt{\frac{10}{7}}}$, $w_1 = \frac{128}{225}$, $w_2 = w_3 = \frac{322 + 13\sqrt{70}}{900}$, $w_4 = w_5 = \frac{322 - 13\sqrt{70}}{900}$.

**Output:** $\frac{b-a}{2}\sum_{i=1}^{5} w_i f(\frac{b-a}{2}x_i + \frac{a+b}{2})$

---

---

**Algorithm 2** Training Process

---

**Input:** Batch size $B$, number of epochs $E$, neural network $f(n; \boldsymbol{\theta})$, training length $N$, maximum extrapolation length $N_{max}$, size of the head $d$.

1: Initialize $\boldsymbol{\theta}$
2: **for** epoch = $1, 2, ..., E$ **do**
3:     Randomly generate $n_1, n_2, ..., n_B$, where $n_i$ is an integer greater than $N$ and less than or equal to $N_{max}$, $i = 1, 2, ..., B$
4:     $\lambda_i \leftarrow \frac{1}{\sqrt{d}} +$ Five-Point Gaussian Quadrature$(N, n_i, f(n; \boldsymbol{\theta}))$, $i = 1, 2, ..., B$
5:     $\nabla_{\boldsymbol{\theta}} L \leftarrow \nabla_{\boldsymbol{\theta}} \frac{1}{B} \sum_{i=1}^{B} \left( H(n_i, \lambda_i) - H(N, \frac{1}{\sqrt{d}}) \right)^2$
6:     $\boldsymbol{\theta} \leftarrow Adam(\boldsymbol{\theta}, \nabla_{\boldsymbol{\theta}} L)$
7: **end for**

---

# 5 EXPERIMENT

We are primarily concerned with two types of problems: Offline 3D Bin Packing Problem(O3BPP) and routing problems. The latter includes the TSP and the Capacitated Vehicle Routing Problem(CVRP).

We present a uniform hyperparameter configuration for three combinatorial optimization problems. The optimization algorithm employed is Adam (Kingma & Ba, 2014), and the Rollout algorithm (Kool et al., 2019) is used for training. When the $p$ value in the Rollout algorithm exceeds 0.95, the learning rate is scaled down by a factor of 0.95. The Rollout algorithm is run for 500 epochs, with 10 steps per epoch, and a significance level of 0.05. The models for all three combinatorial optimization problems consist of multi-head self-attention and feedforward layers. The number of heads $h$ is 8, and the dimension of the head $d$ is 16. There are 3 multi-head attention layers, and the output dimensions of the feedforward layers are 512 and 128, respectively. The temperature coefficient $C$ is assigned a value of 10. LayerNorm (Ba et al., 2016) is used for normalization. $N = 100$, $N_{max} = 1600$, and the test dataset size is 16384.

## 5.1 OFFLINE 3D BIN PACKING PROBLEM

The O3BPP under investigation is defined as follows: given the length, width, and height of $n$ cuboid boxes and a cuboid bin of length $L$ and width $W$, the goal is to pack all $n$ cuboid boxes into the bin such that the bin's height is minimized and the boxes cannot be placed obliquely.

For the O3BPP, we use the attend2pack model (Zhang et al., 2021). In terms of network architecture, we basically follow the original paper (Zhang et al., 2021). The number of convolutional layers is 3, the input channel number of the first convolutional layer is 2, the input channel numbers of the last two convolutional layers are 4, and the output channel numbers of the three convolutional layers are all 4. Their kernel size, stride, and padding are 3, 2, and 1, respectively. All boxes have lengths, widths, and heights that are randomly and equally likely to be integers from 10 to 50. Batch size is 64.

As can be observed from Table 1, our approach outperforms existing methods during extrapolation, especially when $n$ is relatively large, showcasing a significant improvement over the current techniques. Simultaneously, we observe that the performance is quite poor when $\lambda = \frac{\ln n}{\sqrt{d}}$. This can be attributed to an excessively large scale factor during this computation. In the attention calculation process, each box is only able to focus on a small number of other boxes. The study by Chiang & Cholak (2022) also only considers cases where $n$ is relatively small, such as $n = 20$. Henceforth, we will avoid conducting experiments with $\lambda = \frac{\ln n}{\sqrt{d}}$.

## 5.2 TRAVELING SALESMAN PROBLEM

The TSP involves finding the shortest route to visit each of $n$ given cities once and return to the starting city.

Table 1: Experimental results for the O3BPP. The data inside the parentheses in the table represents the standard deviation from five repetitions, while the data preceding the parentheses indicates the average load rate from five repetitions. The larger the load rate, the better. The % symbol has been omitted after the standard deviation and load rate for brevity.

| Test Box Number $n$ | 100 | 200 | 400 | 800 | 1600 |
|---|---|---|---|---|---|
| | | $L = 120, W = 100$ | | | |
| $\lambda = \frac{1}{\sqrt{d}}$ | 78.3(1.3) | 76.5(0.8) | 75.2(1.9) | 71.8(2.9) | 62.6(1.7) |
| $\lambda = \frac{\ln n}{\sqrt{d}}$ | 53.6(1.6) | 54.2(1.5) | 52.0(1.6) | 55.6(2.4) | 50.9(2.2) |
| $\lambda = \frac{\ln n}{d}$ | **78.6**(0.1) | 77.3(1.4) | 76.3(0.3) | 72.9(1.4) | 66.9(0.4) |
| EIE(Ours) | 78.3(1.3) | **78.8**(1.0) | **77.7**(0.7) | **75.0**(1.3) | **72.6**(1.2) |
| | | $L = 140, W = 110$ | | | |
| $\lambda = \frac{1}{\sqrt{d}}$ | **77.6**(1.2) | 75.6(0.4) | 74.0(1.0) | 69.9(2.0) | 62.3(1.1) |
| $\lambda = \frac{\ln n}{\sqrt{d}}$ | 52.3(1.5) | 52.5(1.6) | 52.1(2.7) | 55.8(2.4) | 55.0(2.1) |
| $\lambda = \frac{\ln n}{d}$ | 77.4(1.5) | 76.0(1.2) | 74.4(0.9) | 71.9(0.4) | 66.1(0.6) |
| EIE(Ours) | **77.6**(1.2) | **76.6**(0.6) | **75.9**(0.8) | **73.3**(0.4) | **69.0**(1.7) |
| | | $L = 160, W = 120$ | | | |
| $\lambda = \frac{1}{\sqrt{d}}$ | 76.1(0.3) | 74.1(1.8) | 70.1(1.2) | 69.9(1.7) | 61.2(1.1) |
| $\lambda = \frac{\ln n}{\sqrt{d}}$ | 51.4(2.6) | 55.8(2.5) | 52.4(2.9) | 52.6(2.8) | 55.4(2.9) |
| $\lambda = \frac{\ln n}{d}$ | **76.2**(1.8) | 74.8(0.9) | 72.8(0.2) | 71.0(1.8) | 64.7(1.7) |
| EIE(Ours) | 76.1(0.3) | **75.5**(0.3) | **74.8**(1.1) | **72.6**(0.3) | **70.0**(1.3) |

For the TSP, we utilize the Attention Model[1] (Kool et al., 2019) with an initial learning rate of $10^{-4}$. Batch size is 256. The coordinates of the cities in the training and testing data are randomly generated from a uniform distribution between -1 and 1.

Table 2: The experimental results for the TSP. The number preceding the parenthesis in the table represents the average route length from five repetitions, with lower values being better. The number within the parentheses denotes the standard deviation after five repetitions.

| Test City Number $n$ | 100 | 200 | 400 | 800 | 1600 |
|---|---|---|---|---|---|
| $\lambda = \frac{1}{\sqrt{d}}$ | 18.5(0.7) | 27.5(2.0) | 40.8(2.9) | 61.9(2.2) | 96.6(1.9) |
| $\lambda = \frac{\ln n}{d}$ | **18.4**(0.1) | 27.3(0.8) | 40.5(2.9) | 60.3(1.3) | 96.5(0.4) |
| EIE(Ours) | 18.5(0.7) | **26.0**(1.1) | **39.0**(0.9) | **59.2**(1.6) | **91.2**(4.8) |

Table 2 presents the experimental results of various extrapolation methods for the TSP. Similar to the offline bin packing problem, our method surpasses existing methods, showing a notable improvement over previous methods as the number of cities increases. Additionally, we observe that during extrapolation, $\lambda = \frac{\ln n}{d}$ outperforms $\lambda = \frac{1}{\sqrt{d}}$. This suggests that reducing the relative rate of entropy change indeed enhances the model's extrapolation capability.

---

[1] https://github.com/wouterkool/attention-learn-to-route

## 5.3 Capacitated Vehicle Routing Problem

The CVRP examined in this study refers to a scenario with a given set of coordinates for $n+1$ cities, one of which serves as the origin, and the remaining $n$ cities are destinations. Each destination city has a specific demand for goods, and the origin is equipped with an ample supply of goods and an infinite number of vehicles, each with a capacity of $C_v$. The aim is to utilize these vehicles to cater to the demands of all destination cities, with the smallest total distance covered by all the vehicles being the ultimate goal. The total distance also includes the distance for the vehicles to return to the origin.

For the CVRP, our model and parameters largely align with those used for the TSP. Here, we only present the differences: When $C_v = 15, 20$, training becomes considerably unstable, hence we adjust the initial learning rate to $5 \times 10^{-5}$. The input dimension of the first fully connected layer changes from two to three dimensions. This is because the input for the TSP is two-dimensional, containing only city coordinates, while the input for the CVRP is three-dimensional, encompassing city coordinates and the quantity of goods required by each city. In the training and testing datasets, the quantity of goods needed follows a uniform distribution between 0 and 1.

Table 3: The results of the CVRP experiment. The numbers preceding the parentheses represent the average distance after repeating the experiment five times, with a smaller number being better. The numbers inside the parentheses denote the standard deviation of the distance after repeating the experiment five times.

| Test Destination Number $n$ | 100 | 200 | 400 | 800 | 1600 |
|---|---|---|---|---|---|
| $C_v = 10$ | | | | | |
| $\lambda = \frac{1}{\sqrt{d}}$ | **25.3**(0.2) | 42.7(1.2) | 73.3(0.2) | 126.9(0.5) | 231.7(0.9) |
| $\lambda = \frac{\ln n}{d}$ | 25.3(0.6) | 42.2(1.1) | 71.8(0.8) | 123.8(1.0) | 229.5(0.4) |
| EIE(Ours) | **25.3**(0.2) | **41.2**(0.6) | **69.3**(1.2) | **120.5**(1.3) | **215.0**(1.0) |
| $C_v = 15$ | | | | | |
| $\lambda = \frac{1}{\sqrt{d}}$ | **23.2**(1.2) | 38.6(0.2) | 63.5(0.8) | 106.7(3.2) | 186.3(3.3) |
| $\lambda = \frac{\ln n}{d}$ | 23.9(1.4) | 38.2(1.2) | 62.5(1.3) | 104.9(0.4) | 185.3(1.3) |
| EIE(Ours) | **23.2**(1.2) | **36.7**(0.7) | **60.3**(2.9) | **102.2**(3.9) | **177.9**(2.5) |
| $C_v = 20$ | | | | | |
| $\lambda = \frac{1}{\sqrt{d}}$ | 21.9(0.6) | 38.4(0.4) | 60.6(0.5) | 101.1(1.0) | 169.3(3.4) |
| $\lambda = \frac{\ln n}{d}$ | **21.6**(0.1) | 37.6(0.7) | 58.8(1.3) | 96.3(3.8) | 162.7(0.1) |
| EIE(Ours) | 21.9(0.6) | **34.4**(1.1) | **55.8**(2.6) | **93.0**(1.2) | **159.4**(3.3) |

Table 3 presents the experimental results for the CVRP. The findings further corroborate the superiority of our method over existing methods.

## 5.4 Our Model

Given that in the O3BPP, TSP and CVRP, the training lengths $N$ are all set to 100, and the maximum extrapolated lengths $N_{max}$ are all set to 1600, with a head size $d$ of 16, we only need to train our model once. The number of residual blocks in our model $N_r$ is set to 3, $d' = 10$, with Adam being the optimizer and an initial learning rate of $10^{-3}$. The number of epochs is specified as 500, and after each epoch, the learning rate is multiplied by 0.99. The batch size is set at 256.

Figure 2 illustrates the graph of the learned entropy expectation $H$ against $n$. The graph shows that compared to $\lambda = \frac{1}{\sqrt{d}}$ and $\lambda = \frac{\ln n}{d}$, the relative rate of change in the entropy expectation

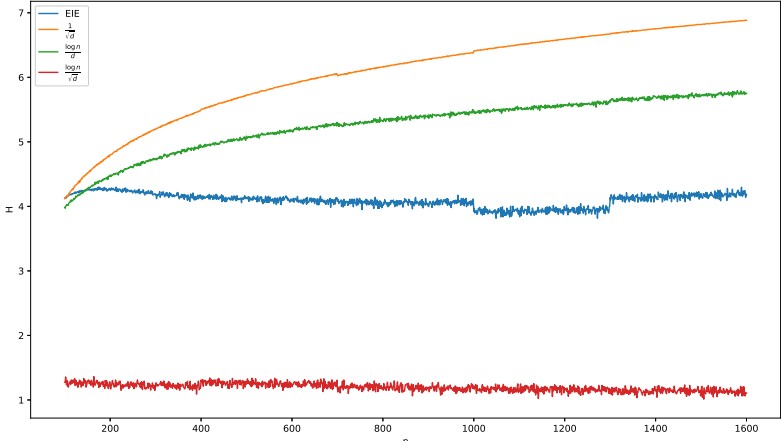

Figure 2: Graph of entropy expectation $H$ against $n$.

for our method is relatively smaller. Concurrently, we observe that when $\lambda = \frac{\ln n}{d}$, the entropy is exceedingly small, which aligns with our prior explanation. From our previous experimental results, $\lambda = \frac{\ln n}{d}$ indeed outperforms $\lambda = \frac{1}{\sqrt{d}}$. The relative rate of change in entropy for $\lambda = \frac{\ln n}{d}$ is indeed less than that for $\lambda = \frac{1}{\sqrt{d}}$. This observation corroborates the validity of our hypothesis.

Table 4 presents the inference times corresponding to different $n$ values. Even though our approach introduces an additional neural network, it only increases the time by 0.02 seconds. This increment is negligible when compared to the original inference time.

Table 4: The inference times of CVRP corresponding to different $n$ values. Our approach increases the time by 0.02 seconds. Please note that the units in the table are in hours.

| $n$ | 200 | 400 | 800 | 1600 |
|---|---|---|---|---|
| $time(h)$ | 0.17 | 0.54 | 2.18 | 8.18 |

## 6   CONCLUSION

In this paper, we have introduced a length extrapolation method for Transformers that does not require position embeddings. This approach is based on the principle of entropy invariance and designs different scaling factors for various extrapolation lengths. Compared to previous works, our method does not necessitate the retraining of the Transformer model, thereby saving a significant amount of time. Our model is also remarkably simple, consisting of just a few hundred parameters. Even with CPU training, the training time is acceptable. Furthermore, extensive experimental results demonstrate that our approach outperforms existing methods.

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

# A APPENDIX

## A.1 PROOF OF THEOREM 1

First, let's establish the following lemma:

**Lemma 1** *If $f : \mathbb{R}^{n \times d_2} \to \mathbb{R}^{n \times d_3}$ and $g : \mathbb{R}^{n \times d_1} \to \mathbb{R}^{n \times d_2}$ are permutation equivariant , then the composition $f \circ g$ also is permutation equivariant .*

**Proof** *Let $\boldsymbol{X} \in \mathbb{R}^{n \times d_1}$ and $\sigma \in S_n$.*

$$f(g(\sigma(\boldsymbol{X}))) = f(\sigma(g(\boldsymbol{X}))) = \sigma(f(g(\boldsymbol{X}))) \tag{14}$$

*Thus, $f \circ g$ is permutation equivariant . □*

Given that the CO-Transformer is composed of multi-head attention layers without positional embeddings, fully connected layers, Softmax, and LayerNorm, if we prove that each individual function is permutation equivariant , then according to Lemma 1, CO-Transformer would also be permutation equivariant. Lee et al. (2019) has already proved that multi-head attention layers without positional embeddings, fully connected layers, and LayerNorm are permutation equivariant. The proof for Theorem 1 is as follows:

**Proof** *We start by proving that the multi-head attention layer is permutation equivariant. We breakdown the computation of the multi-head attention into three types of mappings: $f_i : \boldsymbol{X} \to (\boldsymbol{Q}_i, \boldsymbol{K}_i, \boldsymbol{V}_i)$, $g_i : (\boldsymbol{Q}_i, \boldsymbol{K}_i, \boldsymbol{V}_i) \to \boldsymbol{O}_i$, and $h : (\boldsymbol{O}_1, \boldsymbol{O}_2, ..., \boldsymbol{O}_h) \to \boldsymbol{O}$, where $(\boldsymbol{Q}_i, \boldsymbol{K}_i, \boldsymbol{V}_i) \in \mathbb{R}^{n \times 3d}$. The meaning of the symbols appearing in the above three types of mappings is the same as that in Equations 1, 2, 3 and 4. If we prove that each type of mapping is permutation equivariant , then according to Lemma 1, the multi-head attention layer can be permutation equivariant. We denote the input to the multi-head attention layer as $\boldsymbol{X}$ and any permutation as $\sigma$. Let $\boldsymbol{X}' = \sigma(\boldsymbol{X})$. If the element in the $j$-th row and $k$-th column of $\boldsymbol{X}$ is represented as $X[j, k]$, then the element in the $j$-th row and $k$-th column of $\boldsymbol{X}'$ is represented as $X[\sigma(j), k]$. Mimicking Equation 1 and taking $\boldsymbol{X}'$ as input (any variable with a prime in the superscript indicates the result obtained with $\boldsymbol{X}$ as input), we can get:*

$$\boldsymbol{Q}_i'[j, :] = \boldsymbol{X}[\sigma(j), :]\boldsymbol{W}_Q^i, \boldsymbol{K}_i'[j, :] = \boldsymbol{X}[\sigma(j), :]\boldsymbol{W}_K^i, \boldsymbol{V}_i'[j, :] = \boldsymbol{X}[\sigma(j), :]\boldsymbol{W}_V^i \tag{15}$$

*Here, $i = 1, 2, ..., h$, and $\boldsymbol{X}[\sigma(j), :]$ represents the $\sigma(j)$-th row vector of $\boldsymbol{X}$. Clearly, we have $f_i(\sigma(\boldsymbol{X})) = \sigma(f_i(\boldsymbol{X}))$, so that $f_i : \boldsymbol{X} \to (\boldsymbol{Q}_i, \boldsymbol{K}_i, \boldsymbol{V}_i)$ is permutation equivariant is proven. According to Equation 2, we can get:*

$$\boldsymbol{A}_i'[j, k] = \frac{exp(\boldsymbol{Q}_i[\sigma(j), :]\boldsymbol{K}_i^T[\sigma(k), :]/\sqrt{d})}{\sum_{k=1}^n exp(\boldsymbol{Q}_i[\sigma(j), :]\boldsymbol{K}_i^T[\sigma(k), :]/\sqrt{d})} \tag{16}$$

*According to Equation 3:*

$$\boldsymbol{O}_i'[j, k] = \frac{\sum_{l=1}^n exp(\boldsymbol{Q}_i[\sigma(j), :]\boldsymbol{K}_i^T[\sigma(l), :]/\sqrt{d})\boldsymbol{V}_i[\sigma(l), k]}{\sum_{l=1}^n exp(\boldsymbol{Q}_i[\sigma(j), :]\boldsymbol{K}_i^T[\sigma(l), :]/\sqrt{d})} \tag{17}$$

*With $\boldsymbol{X}$ as the input, $\boldsymbol{O}_i$ is defined as:*

$$\boldsymbol{O}_i[j, k] = \frac{\sum_{l=1}^n exp(\boldsymbol{Q}_i[j, :]\boldsymbol{K}_i^T[l, :]/\sqrt{d})\boldsymbol{V}_i[l, k]}{\sum_{l=1}^n exp(\boldsymbol{Q}_i[j, :]\boldsymbol{K}_i^T[l, :]/\sqrt{d})} \tag{18}$$

*It's clear that $\boldsymbol{O}_i'[j, k] = \boldsymbol{O}_i[\sigma(j), k]$, therefore proving that $g_i : (\boldsymbol{Q}_i, \boldsymbol{K}_i, \boldsymbol{V}_i) \to \boldsymbol{O}_i$ is permutation equivariant. The third type of mapping is similar to the first type, involving only matrix concatenation and multiplication. Similar to the first type, it's straightforward to prove that the third type of mapping $h : (\boldsymbol{O}_1, \boldsymbol{O}_2, ..., \boldsymbol{O}_h) \to \boldsymbol{O}$ also is permutation equivariant . Thus, that multi-head attention is permutation equivariant has been demonstrated. For the fully connected layer, it has been shown that a fully connected layer without Bias and activation function is permutation equivariant , and it's relatively easy to prove that a fully connected layer with Bias and activation function is also permutation equivariant.*

*Next, we proceed to establish that LayerNorm is permutation equivariant.*

As before, we posit that $\boldsymbol{X} \in \mathbb{R}^{n \times (h \times d)}$ is the input to LayerNorm, and $\boldsymbol{Y} \in \mathbb{R}^{n \times (h \times d)}$ is the output of LayerNorm. We define $\boldsymbol{X}' = \sigma(\boldsymbol{X})$, and $\boldsymbol{Y}'$ is the output of LayerNorm when $\boldsymbol{X}'$ is the input.

$$\boldsymbol{Y}'[i,j] = \frac{\boldsymbol{X}'[i,j] - E_j(\boldsymbol{X}'[i,j])}{std_j(\boldsymbol{X}'[i,j]) + \epsilon'} = \frac{\boldsymbol{X}[\sigma(i),j] - E_j(\boldsymbol{X}[\sigma(i),j])}{std_j(\boldsymbol{X}[\sigma(i),j]) + \epsilon'} = \boldsymbol{Y}[\sigma(i),j] \qquad (19)$$

where $E_j(\boldsymbol{X}[i,j]) = \frac{1}{hd}(\sum_{j=1}^{hd} \boldsymbol{X}[i,j])$, $std_j(\boldsymbol{X}[i,j]) = \sqrt{\frac{1}{hd}(\boldsymbol{X}[i,j] - E_j(\boldsymbol{X}[i,j]))^2}$, $\epsilon'$ is a very small positive number.

Lastly, we proceed to demonstrate that Softmax is permutation equivariant. Typically, standalone Softmax is only present in the final layer. Therefore, the input to Softmax can be considered as a vector. We posit that $\boldsymbol{x} \in \mathbb{R}^n$ is the input to Softmax, and $\boldsymbol{y} \in \mathbb{R}^n$ is the output of softmax. We define $\boldsymbol{x}' = \sigma(\boldsymbol{x})$, and $\boldsymbol{y}'$ is the output of Softmax when $\boldsymbol{x}'$ serves as the input.

$$\boldsymbol{y}'[i] = \frac{e^{\boldsymbol{x}'[i]}}{\sum_{i=1}^{n} e^{\boldsymbol{x}'[i]}} = \frac{e^{\boldsymbol{x}[\sigma(i)]}}{\sum_{i=1}^{n} e^{\boldsymbol{x}[\sigma(i)]}} = \boldsymbol{y}[\sigma(i)] \qquad (20)$$

□

## A.2 ADDITIONAL EXPERIMENTAL RESULTS

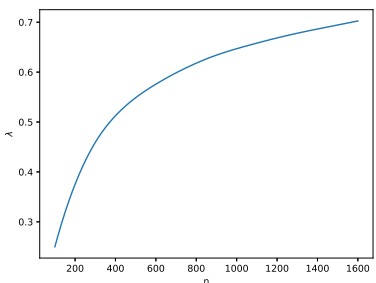

Figure 3: $d = 16$. The function graph of $\lambda$ and $n$ learned from the training.

Table 5: $d = 16$. The values of $\lambda$ corresponding to some particular $n$.

| $n$ | 100 | 200 | 400 | 800 | 1600 |
|---|---|---|---|---|---|
| $\lambda$ | 0.250 | 0.374 | 0.512 | 0.618 | 0.702 |

Figure 3 presents the function graph of the learned $\lambda$ and $n$. Upon examination, the learned function shares some similarities with the logarithmic function, as both exhibit properties of being monotonically increasing functions with monotonically decreasing derivatives. In previous work, the scaling factors all incorporate logarithmic functions. Compared to the standard multi-head attention scaling factor $\frac{1}{\sqrt{d}}$, the function $\frac{\ln n}{d}$ bears more resemblance to the function we learned. This resemblance may explain why $\frac{\ln n}{d}$ tends to perform better than $\frac{1}{\sqrt{d}}$ during extrapolation. Table 5 provides the varying values of $\lambda$ corresponding to different $n$ during our extrapolation process.

For the O3BPP, TSP, and CVRP, we increased the number of multi-head attention layers in the model from 3 to 4, increased the head size $d$ from 16 to 32, increased the sequence length during training $N$ from 100 to 120, and increased the maximum length for extrapolation $N_{max}$ from 1600 to 1920. Table 6 displays the experimental results for $d = 32$, $N = 120$, and $N_{max} = 1920$. Figure 4 shows the graph of $\lambda$ against $n$ for $d = 32$, while Table 7 provides the $\lambda$ values for some specific $n$ values. Figure 5 presents the graph of $H$ against $n$ for $d = 32$.

Table 6: Experimental results for $d = 32$, $N = 120$, and $N_{max} = 1920$

| $n$ | 120 | 240 | 480 | 960 | 1920 |
|---|---|---|---|---|---|
| **O3BPP** | | | | | |
| $L = 120, W = 100$ | | | | | |
| $\lambda = \frac{1}{\sqrt{d}}$ | 77.9(0.9) | 76.0(0.6) | 75.2(1.0) | 71.8(1.4) | 62.7(1.7) |
| $\lambda = \frac{\ln n}{d}$ | **78.3**(0.4) | 76.8(1.2) | 76.0(0.2) | 73.3(1.5) | 66.7(0.3) |
| EIE(Ours) | 77.9(0.9) | **79.1**(1.2) | **77.6**(0.6) | **74.7**(1.2) | **72.4**(1.1) |
| $L = 140, W = 110$ | | | | | |
| $\lambda = \frac{1}{\sqrt{d}}$ | **77.9**(1.3) | 75.4(0.2) | 73.8(0.8) | 69.9(2.1) | 62.3(1.1) |
| $\lambda = \frac{\ln n}{d}$ | 77.2(1.4) | 75.6(1.0) | 74.7(1.1) | 72.0(0.4) | 66.1(0.6) |
| EIE(Ours) | **77.9**(1.3) | **76.7**(0.8) | **75.7**(0.7) | **73.7**(0.6) | **69.4**(2.1) |
| $L = 160, W = 120$ | | | | | |
| $\lambda = \frac{1}{\sqrt{d}}$ | 76.1(0.3) | 74.1(1.8) | 70.1(1.2) | 69.9(1.7) | 61.2(1.0) |
| $\lambda = \frac{\ln n}{d}$ | **76.2**(1.8) | 74.8(0.9) | 72.8(0.2) | 71.0(1.8) | 64.7(1.8) |
| EIE(Ours) | 76.1(0.3) | **75.5**(0.3) | **74.8**(1.1) | **72.6**(0.3) | **70.0**(1.3) |
| **TSP** | | | | | |
| $\lambda = \frac{1}{\sqrt{d}}$ | 20.4(0.9) | 31.8(0.4) | 47.3(0.6) | 69.6(2.5) | 106.6(2.4) |
| $\lambda = \frac{\ln n}{d}$ | **20.3**(0.7) | 30.3(0.6) | 45.6(2.0) | 68.6(0.9) | 104.9(0.8) |
| EIE(Ours) | 20.4(0.9) | **29.9**(0.2) | **44.7**(1.2) | **66.7**(2.1) | **102.9**(1.7) |
| **CVRP** | | | | | |
| $C_v = 10$ | | | | | |
| $\lambda = \frac{1}{\sqrt{d}}$ | **29.2**(0.5) | 50.2(1.9) | 85.9(2.9) | 126.9(0.5) | 254.1(3.9) |
| $\lambda = \frac{\ln n}{d}$ | 30.0(0.0) | 49.8(1.0) | 83.8(0.3) | 147.2(0.3) | 252.8(1.9) |
| EIE(Ours) | **29.2**(0.5) | **48.7**(0.3) | **82.6**(2.2) | **144.7**(1.6) | **247.4**(3.7) |
| $C_v = 15$ | | | | | |
| $\lambda = \frac{1}{\sqrt{d}}$ | **26.5**(0.4) | 44.9(1.3) | 73.8(1.8) | 127.9(1.6) | 210.6(1.7) |
| $\lambda = \frac{\ln n}{d}$ | 26.6(0.3) | 43.1(1.9) | 72.9(1.6) | 125.4(3.5) | 209.9(0.1) |
| EIE(Ours) | **26.5**(0.4) | **42.8**(0.9) | **70.5**(2.1) | **121.5**(3.2) | **204.7**(2.8) |
| $C_v = 20$ | | | | | |
| $\lambda = \frac{1}{\sqrt{d}}$ | **25.1**(0.7) | 41.0(0.5) | 67.8(0.6) | 113.0(2.3) | 186.1(4.5) |
| $\lambda = \frac{\ln n}{d}$ | 25.2(0.2) | 40.1(1.1) | 67.0(1.3) | 111.7(0.5) | 185.1(2.6) |
| EIE(Ours) | **25.1**(0.7) | **39.1**(1.9) | **65.5**(1.8) | **109.0**(2.1) | **182.7**(3.8) |

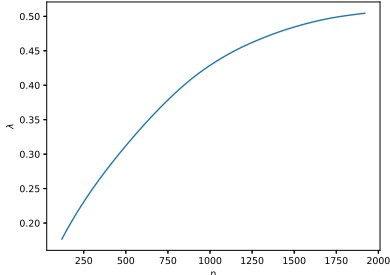

Figure 4: $d = 32$. The function graph of $\lambda$ and $n$ learned from the training.

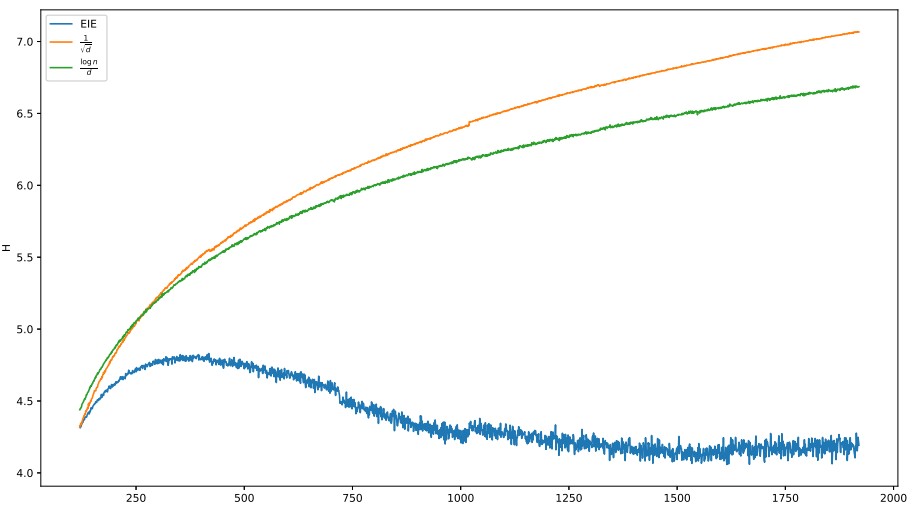

Figure 5: $d = 32$. Graph of entropy expectation $H$ against $n$.

Table 7: $d = 32$. The values of $\lambda$ corresponding to some particular $n$.

| $n$ | 120 | 240 | 480 | 960 | 1920 |
|---|---|---|---|---|---|
| $\lambda$ | 0.177 | 0.227 | 0.306 | 0.422 | 0.504 |

## A.3 RATIONALITY OF ASSUMPTIONS

Figure 6 presents the Jensen–Shannon divergence between each element in $\boldsymbol{X}$ and the standard normal distribution. As can be observed, the Jensen–Shannon divergence for most elements and the standard normal distribution ranges from 8e-3 to 9e-3. This validates our rationale for viewing $\boldsymbol{X}$ as following a standard normal distribution.

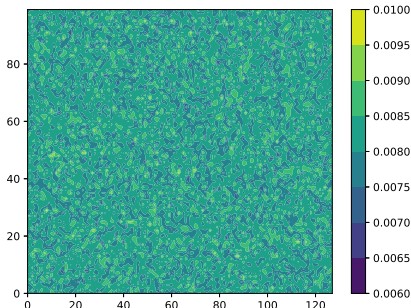

Figure 6: The Jensen–Shannon divergence between each element in $\boldsymbol{X}$ and the standard normal distribution.

Regarding the independence of $\boldsymbol{Q}_{i,:}$ and $\boldsymbol{K}_{j,:}$, if they are not independent, we calculate the entropy using the following formula:

$$
H^{(de)}(n, \lambda(n)) = \mathbb{E}_{\boldsymbol{X} \sim \mathcal{N}(\boldsymbol{0}, \boldsymbol{I}_{n \times hd}), \boldsymbol{W}_Q[:,i], \boldsymbol{W}_K[:,j] \sim \mathcal{N}(\boldsymbol{0}, \frac{1}{hd}\boldsymbol{I}_{hd})}
$$
$$
[\ln \sum_{j=1}^{n} e^{\lambda \boldsymbol{X} \boldsymbol{W}_Q[:,i] \boldsymbol{W}_K[:,j]^T \boldsymbol{X}^T} - \frac{\sum_{j=1}^{n} \lambda \boldsymbol{X} \boldsymbol{W}_Q[:,i] \boldsymbol{W}_K[:,j]^T \boldsymbol{X}^T e^{\lambda \boldsymbol{X} \boldsymbol{W}_Q[:,i] \boldsymbol{W}_K[:,j]^T \boldsymbol{X}^T}}{\sum_{j=1}^{n} e^{\lambda \boldsymbol{X} \boldsymbol{W}_Q[:,i] \boldsymbol{W}_K[:,j]^T \boldsymbol{X}^T}}] \quad (21)
$$

Figure 7 illustrates the entropy calculated using Equations 6 and 21. From the figure, we can discern that the independence (or lack thereof) between $\boldsymbol{Q}_{i,:}$ and $\boldsymbol{K}_{j,:}$ has a negligible impact on the entropy calculation.

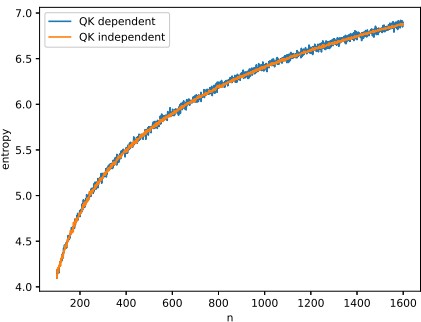

Figure 7: The entropy calculated using Equations 6 and 21.

