# OpenReview forum: "Train Short, Test Long In Combinatorial Optimization"
_ICLR.cc/2024/Conference — ICLR 2024 Conference Withdrawn Submission_

### Official Review · Reviewer_edwp · 2023-10-18

**Soundness:** 2 fair
**Presentation:** 2 fair
**Contribution:** 1 poor
**Rating:** 1
**Confidence:** 4

**Summary:**

The paper tackles combinatorial optimization (CO) tasks using a transformer. The transformer, that according to theorem 1 is permutation equivariant, is used inside a simple algorithm (see Alg. 2) for tackling CO. The intuition is that the model can be learnt using sequences of a certain length, but then be tested using larger sequences and it should still work. The paper provides an intuition based on the entropy on how this can happen and formulates the objective following this intuition. The paper conducts experiments in 3d bin packing, traveling salesman problem (TSP), and capacitated vehicle routing problem.

**Strengths:**

+ CO tasks are important in theoretical CS and have diverse applications.

+ The generalization in CO tasks beyond the instances the models are trained on is a significant challenge.

**Weaknesses:**

The paper presents a large number of weaknesses, which largely outweigh the benefits.

- There is **no related work on CO**, while there are multiple papers on this topic the last few years. A more proper search is required to include and discuss the related work. To provide concrete references, the popular “Pointer networks” (Vinyals et al) and the papers citing it, or the thorough surveys of a) Machine Learning for Combinatorial Optimization: a Methodological Tour d'Horizon (Bengio et al), b) Combinatorial Optimization and Reasoning with Graph Neural Networks (Cappart et al), c) End-to-end constrained optimization learning: A survey (Kotary et al) provide a good starting point. The lack of related work makes it unclear where the paper is placed with respect to the literature.

- Theorem 1 is incomplete. For instance, neither any definition, the theorem or the proof refer to the exact formulation of the transformer that is being “proved”. In addition, the proof does not offer an explanation of why each component (e.g. softmax, layer norm) are permutation equivariant.

- The paper requires thorough proofreading, as it currently is not a format ready to be accepted by top-venues, such as ICLR: a) “Xavier initialization” (versus “Xaiver”), b) “are independent of n, Thus”, c) “As such, We propose”, etc.

- The experimental validation is unclear and non-standard, which does not make it easy for the reader to understand. The vehicle routing and the TSP are some problems that are commonly met, but a reference to the literature would be helpful for the readers that want to find out more. However, in all tasks, there are no comparisons with other methods (see the lack of literature mentioned previously). There are decades of literature in those tasks, with heuristics and/or deep learning methods developed. **The lack of comparisons makes it impossible to evaluate this method**.

**Questions:**

- What is a “box” mentioned in sec. 4?

- Is there any proof of the claim “a well-trained transformer is accustomed to a certain range of entropy”? Could the authors elaborate on that?

- The conclusion mentions that the model has only a few hundred parameters, however there is no table referring to the parameters or the runtime comparison of the proposed models. Could the authors elaborate on that?

- I am wondering why those specific tasks are selected. Does this approach extend to other tasks?

- What are the exact contributions with respect to the Press et al, ICLR’21 paper?

---

> ### Author Response · Authors · 2023-11-19
>
> Thank you for your insightful comments and valuable suggestions. We have uploaded our revised edition.
>
> **No related work on CO.**
>
> In the revised version, the first paragraph of Section 1 introduces the connection between Combinatorial Optimization (CO) and Machine Learning (ML), along with some advancements in the combination of CO and ML.
>
> **Theorem 1 is incomplete, and the proof does not offer an explanation of why each component.**
>
> In our revision, we have defined the CO-Transformer to distinguish it from the Transformer in natural language processing(NLP). We have introduced the components of the CO-Transformer. Although we still do not provide formulas for each component, we have added references to indicate that the formulas for each module are consistent with those in these references. For the proof, we have supplemented it with demonstrations that LayerNorm and Softmax are permutation equivariant.
>
> **The paper requires thorough proofreading.**
>
> We apologize for our oversight. In the revised version, we have made an effort to correct these grammar and spelling errors.
>
> **The experimental validation is unclear and non-standard.**
>
> We deeply respect the achievements of researchers in the field of combinatorial optimization. However, our paper primarily focuses on how to improve the length generalization capability of existing CO-Transformers in combinatorial optimization tasks. Heuristic methods and CO-Transformers are not directly related. Some deep learning methods used in our experiments are our baseline models, i.e., models where $\lambda$ remains constant, and others are not CO-Transformers. You can refer to [1], which studies how to improve the length generalization ability of Transformers in natural language processing. It also only compares with works related to length generalization, excluding other models. To avoid various factors affecting experiment results, many related experiments aiming to enhance Transformer's length generalization ability even fix the model structure to ensure only one variable during the experimentation.
>
> **What is a “box” mentioned in sec. 4?**
>
> We apologize for the oversight. We wanted to use the bin packing problem (BPP) as an example but overlooked the need to explain BPP initially. In the revised version's Section 4.1, we have added an explanation for BPP.
>
> **Is there any proof of the claim “a well-trained transformer is accustomed to a certain range of entropy”? Could the authors elaborate on that?**
>
> We apologize for the confusion. This is not a rigorous theory but a hypothesis we proposed. For instance, in offline BPP, we once trained a model with 50 boxes, which we refer to as Model A, and another model with 100 boxes, which we refer to as Model B. The performance declined whether we tested Model A with a 100-box test set or Model B with a 50-box test set. Hence, we conjecture that it may be related to entropy.
>
> **The conclusion mentions that the model has only a few hundred parameters, however there is no table referring to the parameters or the runtime comparison of the proposed models. Could the authors elaborate on that?**
>
> In the revised version, Table 4 lists the inference time and shows that our method only adds an additional 0.02s.
>
> **I am wondering why those specific tasks are selected. Does this approach extend to other tasks?**
>
> Tasks such as TSP, CVRP, and O3BPP involve sequential decision-making. TSP and CVRP concern the order of visiting locations, while O3BPP deals with the order of selecting boxes. These order-related decisions require the neural network to be permutation-equivariant, hence the use of self-attention in neural networks to solve these problems. Our method requires the neural network to contain self-attention. This approach can be extended to other sequential decision-making combinatorial optimization problems, such as the Job Shop Scheduling Problem (JSSP), where the order of jobs needs to be determined.
>
> **What are the exact contributions with respect to the Press et al, ICLR’21 paper?**
>
> [2] designed a new position embedding to improve the Transformer's length generalization ability, and the experiments in [2] are only related to NLP. However, it is well-known that introducing position embeddings makes CO-Transformers non-permutation equivariant, rendering the work of [2] unsuitable for the field of combinatorial optimization. In contrast, our work modifies the scaling factor while maintaining the model as permutation equivariant, making our contributions applicable to the field of combinatorial optimization.
>
> [1]The Impact of Positional Encoding on Length Generalization in Transformers.
>
> [2]Train Short, Test Long: Attention with Linear Biases Enables Input Length Extrapolation

---

> > ### Comment · Reviewer_edwp · 2023-11-19
> > **Not convinced by the response**
> >
> > Dear authors,
> >
> > thank you for your responses to the questions. I am unfortunately not convinced though, since many of my original points still stand. For instance, I mentioned explicitly that there is a lack of comparisons (despite of several decades of research on this topic) and yet the paper still does not include any comparisons.

---

> > > ### Author Response · Authors · 2023-11-20
> > >
> > > Thank you for your response. We would like to reiterate that our research objective is to improve the length generalization ability of the existing CO-Transformer. Our experiments are conducted with controlled variables and other means to enhance the length generalization ability of the CO-Transformer. The reviewer emphasizes that there are decades of research in the field of combinatorial optimization. Does the reviewer view Natural Language Processing (NLP) as a new discipline? Does the reviewer believe that [1] published in Neurips 2023 should be compared with classic models like LSTM?
> > >
> > >
> > > [1]The Impact of Positional Encoding on Length Generalization in Transformers.

---

### Official Review · Reviewer_nV4X · 2023-10-29

**Soundness:** 2 fair
**Presentation:** 1 poor
**Contribution:** 1 poor
**Rating:** 3
**Confidence:** 4

**Summary:**

This paper studies "length" extrapolation in the context of Transformer-based combinatorial optimization solvers. The authors propose an entropy invariant extrapolation method called EIE to improve the extrapolation ability of Transformers. Specifically, EIE learns the derivative of the temperature in attention modules. The authors show that EIE leads to some performance gains experimentally.

**Strengths:**

- It's important to study extrapolation for neural-network-based combinatorial optimization solvers.

- The proposed method shows some performance improvement in the experiments (although it's unclear how much computational overhead it brings).

**Weaknesses:**

**Related works are significantly under-discussed and under-cited.**
- In Sec. 1, relevant papers should have been cited to support the claims (e.g., the significance of combinatorial optimization problems, failures of traditional solvers, the emergence and success of neural-network-based solvers, etc).
- Sec. 2 misses many related papers. To name a few:
   - [1] Location Attention for Extrapolation to Longer Sequences.
   - [2] Monotonic Location Attention for Length Generalization.
   - [3] Induced Natural Language Rationales and Interleaved Markup Tokens Enable Extrapolation in Large Language Models.
   - [4] From Local Structures to Size Generalization in Graph Neural Networks.
   - [5] Conditional Positional Encodings for Vision Transformers.
   - [6] The Impact of Positional Encoding on Length Generalization in Transformers.

**Factual errors.**
- Def. 1 is called _permutation equivariant_ in literature.
- Thm. 1 has to specify "Transformer _encoders_", since Transformer decoders are not permutation equivariant.


**Problematic assumptions.**
- In Sec. 4.1, each element in $\mathbf{X}$ is assumed to be Gaussian without enough justification.
- Furthermore, in Equation 6, it's claimed that $\\mathbf{Q}\_{i,:}$ and $\\mathbf{K}\_{j,:}$ are Gaussian, but their joint distribution is not specified. I check the code and it seems that the authors implicitly assume $\\mathbf{Q}\_{i,:}$ and $\\mathbf{K}_{j,:}$ are independent because of these two lines:
```
    q=torch.randn(batch_size,1,dim)
    y=torch.randn(batch_size,dim,N)
```
$\qquad$This is problematic to me because $\mathbf{Q}=\mathbf{XW}_Q$ and $\mathbf{K}=\mathbf{XW}_K$ are generally not independent by definition (unless $\mathbf{W}_Q$ and $\mathbf{W}_K$ satisfy some strong assumptions).

**Weak evaluation.**
- The authors should have also discussed the efficiency of the proposed method, as it introduces an additional neural network.

**Presentation issues.**
- Sec. 4.1, "the entropy of the i-th row": i should be italic.
- Redundant "," in Equations 1-3.
- In Figure 2, legend and numbers are too small for readers. The caption is very unclear.

**Questions:**

See **weaknesses**.

---

> ### Author Response · Authors · 2023-11-19
>
> We sincerely thank you for taking the time to review our paper and provide constructive suggestions. We have since updated and submitted a revised version.
>
> **Missing related work:**
>
> In the revised version, we have supplemented the first paragraph of Section 1 with literature on the combination of machine learning and combinatorial optimization. In Section 2, we have incorporated the references you pointed out, with the exception of [1], which studies the generalization of nodes in graph neural networks.
>
> **Factual errors:**
>
> We appreciate your keen eye for detail. In the revised manuscript, we have corrected the factual errors you indicated. The term "permutation equivariant" is now the only one used. Given that the Transformer used in combinatorial optimization is not entirely equivalent to the Transformer encoder [2], we have introduced a new term, "CO-Transformer."
>
> **Problematic assumptions:**
>
> In the updated manuscript, Figure 6 in Section A.3 presents the JS divergence of each element in X from the standard normal distribution. Figure 7 compares the entropy calculated under the assumption that Q, K are independent versus when they are not.
>
> **Weak evaluation:**
>
> In the revised version, Table 4 lists the inference time and shows that our method only adds an additional 0.02s.
>
> **Presentation issues:**
>
> We have made some adjustments in the revised version for better clarity. The variable $i$ is now italicized in Section 4.1, and redundant commas in Equations 1-3 have been removed. The original Figure 2 has been split into two separate figures, now numbered as Figures 2 and 3, both of which have been enlarged for better visibility.
>
>
> [1]  From Local Structures to Size Generalization in Graph Neural Networks.
>
> [2] Attention Is All You Need.

---

### Official Review · Reviewer_zexN · 2023-10-31

**Soundness:** 2 fair
**Presentation:** 2 fair
**Contribution:** 2 fair
**Rating:** 5
**Confidence:** 3

**Summary:**

The authors propose a way to adapt pretrained permutation invariant transformers trained to solve combinatorial problems to longer sequences. The method does not require any re-training of the underlying transformer model.

**Strengths:**

- Combinatorial problems are an interesting application of permutation invariant transformers
- The results are impressive and show an increase on longer sequence length problems without requiring retraining.

**Weaknesses:**

- Theorem 1 has been proposed in previous works. I believe the first work was [1]. This should be cited as such in the text.

- The assumption that the inputs after layernorm are Gaussian may be too strong of an assumption. For instance, if the input experiences a feature collapse, then all the values could collapse to a constant even after layernorm. While this is unlikely, it would be more reassuring to present some empirical result of the distributions of the features after layernorm on the models which are already trained.

- I do not understand the statement below equation 8. You state that you need to ensure the derivative is positive and it would be difficult to ensure that equation 8 holds if you were to use a neural network to directly approximate $\lambda$. In order to enforce the positivity condition, you apply a non-negative activation function to the derivative approximation network. Couldn't this same non-negative function be used on a network to directly approximate $\lambda$? If you were to do this and check every $n$ in the integration, you would find that they are all positive, which means that the positivity condition has been met.

- I do not understand the justification for such a complex opertaion to integrate and train the model (algorithms which are presented in Algorithm 1 and 2). Building on the point above, if $\lambda$ could be predicted directly, the process would become much simpler. If indeed $\lambda$ cannot be predicted directly, then it would be good to include an ablation study of what happens when it is attempted to be directly predicted, the outcome of this study would validate the hypothesis presented about predicting the derivative.

- Figure 2 has an almost meaningless caption which doesn't describe the figure well.

- The conclusion states that the model contains only a few hundred parameters, and that training time can be done on a CPU. Doesn't the entropy calculation on line 5 of Algorithm 2 require the inputs in order to train? If so, then the underlying transformer model must be run in order to calculate equation 6. Even if it does not require tracking gradients for the underlying model, it seems like this should be mentioned, because it could potentially be a huge transformer.

**Questions:**

- At the end of section 2, it is stated that "In the aforementioned work, the logarithm log is presumed to have the natural number e as its base." Why not just use $\ln n$ in the expression if this is the case?

- Why is it that other experiments which utilize permutation invariant transformers do not suffer from degrading performance when extrapolating to longer sequences during testing? For instance, in [2] (Table 1), the performance of a permutation invariant transformer sees an increase in performance when drastically increasing the set size during testing. Do you have any intuition why this might be the case even though it no doubt exhibits the same increase in entropy in the attention matrix?

### References
 - [1] Set Transformer - https://arxiv.org/pdf/1810.00825.pdf

 - [2] Slot Set Encoder - https://arxiv.org/pdf/2103.01615.pdf

---

Overall, I think the results presented are interesting, but I do not see a clear justification of the proposed method (see points above in "Weaknesses"). If the authors can present a concrete reason why the integration is needed (with an ablation study if possible), I would consider raising my score.

---

> ### Author Response · Authors · 2023-11-19
>
> We greatly appreciate you taking the time to review our paper and for your valuable feedback. We have since uploaded a revised version.
>
> **Theorem 1 has been proposed in previous works.**
>
> Thank you for pointing this out. In the proof of the revised version, we have referred to [1].
>
> **The assumption that the inputs after layernorm are Gaussian may be too strong.**
>
> In the revised version, Figure 6 in sec A.3 showcases the JS divergence of each element from the standard normal distribution.
>
> **Ablation study about directly approximating $\lambda$.**
>
> We conducted an ablation study for this with the metric chosen as $\frac {1}{N_{max}-N} \sum_{n=N+1}^{N_{max}} (H(n,\lambda(n)-H(N,\frac {1}{\sqrt{d}}))^2$.
>
> Here, the metric for not directly approximating $\lambda$ is 7.6e-3, while that for directly approximating $\lambda$ is 1.2e-2. Currently, it seems that not directly approximating $\lambda$ delivers better results.
>
> **Figure 2 has an almost meaningless caption which doesn't describe the figure well:**
>
> In the revised version, the original Figure 2 has been renumbered as Figures 2 and 3.
>
> **Doesn't the entropy calculation on line 5 of Algorithm 2 require the inputs in order to train?**
>
> As of now, EIE does not require the outputs of the underlying Transformer. We made an assumption that Query and Key are independent. You can refer to Figure 7 in Sec A.3 for the validity of this assumption.
>
> **Why not just use $\ln n$ in the expression?**
>
> We appreciate your suggestion and have now adopted $\ln n$ across the board.
>
> **Question 2**
>
> Our understanding of the set classification problem is limited, and we welcome any corrections if our interpretation is off. As we understand it, set classification is similar to image classification in that it involves a one-step classification; whereas, combinatorial optimization problem is akin to the autoregressive model in NLP, involving multiple step classifications, where the result of the previous step impacts the next. Consequently, errors accumulate over time, significantly affecting the final result.
>
>
> [1]  Set Transformer: A Framework for Attention-based Permutation-Invariant Neural Networks.

---

### Official Review · Reviewer_SN8V · 2023-10-31

**Soundness:** 2 fair
**Presentation:** 2 fair
**Contribution:** 2 fair
**Rating:** 5
**Confidence:** 3

**Summary:**

In this paper the authors propose an adaptation strategy for making transformer models to combinatorial optimization problems. Essentially, the key element of novelty resides in the "entropy invariant extrapolation" method (EIE). This builds on some adaptations for making transformer models work on these problems:

- permutation invariance should be guaranteed;
- the entropy of the output for multi-head attention should be as low as possible (given that it passes through some softmax, such quantity is bound);
- an auxiliary neural network is trained to tune the inverse temperature in the softmax, which tunes as well the entropy.

The network architecture design relies on ReQUr and ReQU activations. The experiments are conducted on some problems of combinatorial optimization, and the comparison is performed against other fixed inverse temperature scalings.

**Strengths:**

- the employment of transformers for combinatorial optimization is interesting, more specifically the employment of a transformer-like architecture seems intriguing
- the motivation behind permutation invariance is very clear and further motivated in the appendix
- the experiments are averaged on 5 different seeds

**Weaknesses:**

- the experimental section is very limited
- the text is in many points difficult to follow (eg. in (4) it is unclear what $(O_1, O_2,..., O_h)$ is - is it a concatenation? Or rather, is the "box" definition in Sec.4.1 is never provided)
- Sec. 2 is essentially a "list" of works for IPO with not much connection, and the discussion around NPO is very limited
-  there is no ablation study regarding the designed approach
- in most of the experiments, the proposed approach is within 1std other fixed $\lambda$ approaches (which do not involve the optimization of $\lambda$ and for this reason are computationally less expensive). For this, the proposed approach does not seem very effective
- the quality of the figures is not great - for example, Fig.2b requires massive zoom, and the captions are in general not well-explanatory

**Questions:**

- Can you compare EIE to different annealing policies for $\lambda$ (exponential, linear)?
- Can you provide a study as a function of the depth of the employed transformer model?
- Can you elaborate more on why in Fig.2b the red curve behaves better than the blue, and yet is outperformed?

---

> ### Author Response · Authors · 2023-11-19
>
> Thank you very much for taking the time to review our paper and providing valuable suggestions. We have uploaded a revised version.
>
> **The text is in many points difficult to follow:**
>
> We apologize for any confusion. Indeed, eq(4) represents concatenation; "box" in sec 4.1 can be considered as a "token" in language. We initially intended to use bin packing problem as an example, but overlooked the need to explain it beforehand. In the revised version, we have added "concat" in eq (4) and clarified the bin packing problem in sec 4.1.
>
> **The discussion around NPO is very limited:**
>
> We apologize for the limited discussion around the NPO. We would’ve liked to delve deeper into NPO works, but there are not many available.
>
> **No ablation study:**
>
> From our understanding, the ablation study corresponds to keeping $\lambda$ constant.
>
> **The proposed approach does not seem very effective:**
>
> In the revised version, Table 4 presents the inference time for all samples using our method. The inference time only increases by 0.02s.
>
> **The quality of the figures is not great:**
>
> In the revised version, we have enlarged Figure 2b, now labelled as Figure 2.
>
> **Compare EIE to different annealing policies and why in Fig.2b the red curve behaves better than the blue, and yet is outperformed:**
>
> As per your suggestion, we compared our method to linear and exponential functions, $\lambda=\frac{n}{N\sqrt{d}}$ and $\lambda=\frac{e^{n/N-1}}{\sqrt{d}}$, using the boxing problem as an example.
>
> | |  200 | 400|
> |---|---|---|
> |EIE| 78.8 | 77.7|
> |Linear| 52.4 | 51.3|
> |Exp | 51.5 | 50.7 |
>
> The results show that both linear and exponential functions do not perform well. To address your question about why the red curve performs worse in Figure 2 (Figure 2b in the original paper), it is due to the excessively large $\lambda$, which easily leads to one-hot after softmax, resulting in a small gradient. This is also mentioned in [1].
>
> **Can you provide a study as a function of the depth of the employed transformer model?**
>
> We apologize for not fully understanding your suggestion. Are you asking for literature on the approximation capabilities of various activation functions in the transformer model? If so, we regret to inform you that we do not have such resources. The activation functions in our transformer are all ReLU. The reason for using ReQU and ReQUr in EIE is that they perform better than ReLU in approximating smooth functions when the model only contains fully connected layers.
>
> [1]  Attention Is All You Need.